# The Impacts of Urbanisation on Landscape and Environment: The Case of Slovakia

**Zita Izakovičová** [1,2], **František Petrovič** [2] and **Eva Pauditšová** [3,4,*]

1  Institute of Landscape Ecology, Slovak Academy of Sciences, Štefánikova 3, 811 06 Bratislava, Slovakia;
   zita.izakovicova@savba.sk
2  Department of Ecology and Environmental Sciences, Constantine the Philosopher University in Nitra, Tr. A.
   Hlinku 1, 949 01 Nitra, Slovakia; fpetrovic@ukf.sk
3  Department of Environmental Ecology and Landscape Management, Faculty of Natural Sciences, Comenius
   University in Bratislava, Ilkovičova 6, 842 15 Bratislava, Slovakia
4  Institute of Management, Slovak University of Technology, Vazovova 5, 812 43 Bratislava, Slovakia
*  Correspondence: eva.pauditsova@stuba.sk

**Abstract:** The development of urbanisation is linked to qualitative and quantitative changes in the landscape and its components aimed at strengthening economic, administrative and cultural-social functions, which are associated with ever-increasing pressures on ecosystems and their individual components. These pressures are subject to various factors—socio-economic, political, environmental, etc. In this paper, we present an evaluation of the environmental impacts of the development of urbanisation in Slovakia. Independent Slovakia belongs to the young European states. The communist period lasted from 1948 to 1989. The character of landscape, the quality of the environment and also the degree of anthropisation of territory were dependent on a centrally managed economy. Urbanisation changes began to manifest themselves rapidly after 2000, when the economy was transformed and Slovakia was preparing to join the European Union (Slovakia joined the European Union in 2004). The transformation from central planning into a market economy was the basis of the changes, which was conditioned following strong pressure of investors on the landscape, the construction of technological parks, shopping and logistics centres and transport infrastructure and the construction of residential complexes. According to the European Environment Agency's study on urban sprawl between the mid-1950s and the end of the 1990s, industry, commercial and transport services have grown at a significant rate and the residential areas at a moderate rate in Slovakia. On the other hand, Slovakia has areas where urbanisation has the opposite trend. Rural settlements are abandoned as well as large areas of agricultural land. The character of land use has fundamentally changed over the past 20 years. These changes not only have a spatial dimension but are associated with the emergence of various environmental problems. The paper deals with the impacts of anthropisation and industrialisation of Slovakia after 2000. The anthropisation process in Slovakia was determined through data processed in GIS and also through the statistical data representing land use. Based on the ecological significance of land use elements, the degree of anthropisation in Slovakiawas calculated.

**Keywords:** urbanisation; landscape; anthropisation; built-up area

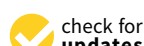

## 1. Introduction

Independent Slovakia belongs to the young European states. The communist period lasted from 1948 to 1989 in the different political-administrative forms. The nature of the landscape was dependent on a centrally managed economy. The forms of land use were influenced mainly by activities related to collective agriculture and industry, as well as urbanisation processes. Until the process of transformation, urbanisation was centrally managed. The development of only certain settlements, the so-called central ones, were preferred which developed rapidly, and on the other hand, the remaining settlements were slowly declining. The construction of houses and flats was regulated centrally, with

precisely set standards—certain types of houses were prescribed which was reflected in a certain uniformity of the settlements, thus the typical monotonous "socialist" towns and villages were created. This development lasted until 1989—the Velvet revolution when a significant transformation of society began. Eastern Europe until today has been carrying many communist-related features, structures and procedures, represented by a variety of landscape icons [1].

Transformational changes not only in Slovakia but also in the other former countries of the socialist and communist bloc significantly influenced the development of urbanisation and suburbanisation, too. There has been a significant spatial development not only of towns but also of rural settlements, especially those that were located near the towns and were easily accessible. On the other hand, rural settlements with poorer accessibility and insufficient infrastructure remain abandoned and often turned into holiday homes. These are mainly villages located in the marginal areas of Slovakia. Of course, the arrival of capital and the market as well as a change of ownership has significantly transformed Eastern European towns and villages. In the process of transformation, there was significant pressure from investors to build technology parks, shopping and logistics centres, transport infrastructure and also the construction of housing complexes and the necessary infrastructure. Spontaneous development, various land use transformations, new shopping centres, industrial parks and logistics centres, emerging neighbourhoods and recreational areas: all of this has profoundly changed not only their functionality but also the cities' look, [2]. Currently, the intensive development of some functions (housing, services, industry) contributes to the weakening of other functions (arable area, afforestation). Additionally, we can observe the phenomenon of the so-called 'fight for space', i.e., gaining as much space as possible to meet our needs [3]. At present in Europe, despite the reduced rate of population growth [3–7], the literature indicates a significant increase in the number of households due to the trend towards smaller households, especially in cities [8–10]. The increase in the number of households is related to an increase in land consumption [11] due to the additional demand for land to house new housing. It should be noted that the space is not only a bare area but also represents the occurrence of natural resources with certain given properties.

The conversion of arable land into building land for various investment projects represents activities with a significant negative environmental impact, such as the liquidation of valuable habitats and ecosystems due to the reduction of forest and settlement vegetation, fragmentation of the landscape, the use of the best soils for non-agricultural activities, etc. These trends have had a negative effect not only on the change in ecosystems and threats to their ecosystem services [12] but also on the deterioration of the quality of the environment. By increasing the degree of development, the addition of new technical buildings, the degree of anthropisation of the area is constantly growing, which, in turn, also negatively affects climate change [13]. The anthropisation of an area creates conditions for increasing the negative impacts of climate change, especially in settlements with a high population density, a high proportion of built-up areas, impermeable surfaces and a high concentration of economic activity and infrastructure. In cities, the effect of the heat island is intensifying, there are heat waves, the number of tropical days and nights is increasing, the infrastructure in cities does not have sufficient capacity for high volumes of water runoff during heavy rainfall and the incidence of floods is increasing. The cities of the future must also prepare for the more frequent occurrence of extreme weather situations [14,15]. New buildings and their areas occupy the landscape intensively.

Urban growth, suburbanisation and the change in the image of "socialist" settlements can be considered as basic urbanisation processes not only in Slovakia but also in other post-socialist European countries, especially in Central Europe [16–22]. This development is also linked to various environmental and socio-economic problems. Several works are devoted to the evaluation of socio-economic problems associated with urbanisation [20,23–25]. The issue of assessing the environmental impacts of urban development in European towns is

less elaborated [26,27]. Many more American authors evaluate the environmental problems associated with the development of urbanisation [28–31].

The main goal of this work is to evaluate industrialisation and urbanisation trends in Slovakia in the transformation period and identify their negative impacts associated with increased space requirements. Given that space represents the integration of natural resources in a given area, its scope has much greater negative environmental consequences. The paper also focuses on the assessment of environmental problems related to the use of natural resources in the area (understanding the landscape as a geosystem).

These urban changes and their impacts pose a challenge for spatial planning and urban development, which is often not sufficiently coordinated and aligned with the potential of the territory [28]. Economic interests that need to be coordinated with environmental interests dominate the development of urbanisation. Effective spatial planning based on the acceptance of landscape ecological restrictions, while respecting the current needs and requirements of society, e.g., building adaptation measures to mitigate the impacts of climate change [32], will significantly reduce the negative environmental impacts of urbanisation processes [33].

Hypotheses:

- The transformation process in Slovakia caused a rapid growth of industrialisation and urbanisation, which was subsequently reflected in the growth of anthropisation of the territory and creation of brownfields of new anthropogenic objects built on areas which were originally arable land or vegetation areas (green areas).
- The development of industrialisation, urbanisation and suburbanisation changes the image and nature of the traditional landscape, and it also reduces the visibility of many attractive and rare landscape types—many historical landscape structures are endangered and disappearing.
- The development of urbanisation is connected with the threat to natural resources and also to a loss of biodiversity—reducing the share of natural ecosystems, gene pool areas, occupation of quality soils and reducing green areas, etc.
- The increase in built-up areas is not conditioned by population growth, but rather is a consequence of a change in lifestyle.

## 2. Materials and Methods

The basis of the methodological procedure was the assessment of changes in land use due to the development of urbanisation and their negative impacts on the landscape and its individual landscape components. To assess the spatial changes in land use elements, we used GIS tools and the Corine Land Cover and Land Cover Change databases which map the state and changes in the land cover of European countries using remote sensing methods from satellite images and national-level database resources [33]. Data-based processes on regular and quantitative inventories of the Copernicus Corine Land Cover, Urban Atlas and Imperviousness data sets, using medium- and high-resolution remote sensing images provide the base for assessment of urbanisation due to changes in types of land cover in all European countries [34].

We used the database for the years 2000 and 2018 [35] to evaluate the changes. In the assessment, we focused not only on the spatial growth of urbanized spaces but we also evaluated the occupation of individual land use elements for the development of urbanisation. The period between 1990 and 2000 was not characterised by a construction boom for Slovakia. It was the period when the transformation of the economy took place, and many manufacturing enterprises, whose history dates back to the times of socialism, disappeared. After 2000, large-scale constructions began to increase rapidly, first being logistics centres and big shopping centres and, later, huge industrial parks. For a more detailed assessment of the development of towns and their functional areas, we also used data from the Urban Atlas, which provides comparable pan-European land cover and land use data for Functional Urban Areas [36]. GIS tools have been also applied for the

spatial identification of conflicts of interest (loading of various map layers) arising from urbanisation and industrialisation.

Subsequently, we supplemented the spatial assessment of changes in urbanisation with more detailed statistical assessments. We used the Statistical Yearbooks of the Slovak Republic [37,38] and also the departmental statistics—Statistical Yearbook and the Land Fund of Slovakia [39], forest data from the National Forest Centre. We have assessed the use of the area and its changes for selected years according to the types of land. We focused on the assessment of changes in the 8 basic types of land that are most affected by the development of urbanisation, namely: arable land, vineyards, orchards, gardens, permanent grassland, forest areas, built-up areas and other areas. Based on the representation of individual elements of land use/corine in spatial units and their landscape ecological significance, the degree of anthropisation of the territory was recalculated. There are several coefficients to evaluate the impact of human activities on landscape. Complexly, the evaluation coefficients are elaborated in the work Landscape ecology [40]. There are also various GIS tools for calculating landscape indicators—StraKa (Landscape Structure) is a GIS tool developed for the analysis of landscape structure. It presents algorithmic and programmed solutions for a set of complex formulas published in summary by Forman and Godron [40]. V-LATE (Vector-based Landscape Analysis Tools Extension) provides a set of the most used so-called metric functions (landscape-ecological indices) to study and determine the landscape structure, etc. [41]. In Slovakia, the assessment of the degree of anthropisation is the most widely used coefficients of ecological stability Buček, Lacina & Löw [42], Miklós [43], Reháčková, Pauditšová [44], etc. In our assessment, we used the coefficient of ecological stability according to Miklós [43] as it assesses in detail the landscape-ecological significance of land use elements relevant to the territory of Slovakia:

$$K_{es} = \sum_{i=1}^{n} p_i \cdot k_{pi} / P$$

where $K_{es}$ is the coefficient of quality of the spatial structure of the territory, $p_i$ is the area of individual elements in hectares, $k_{pi}$ is the coefficient of landscape ecological significance of elements, P is the area in hectares and n is the number of elements in the area. In total, we considered 8 elements of land use: forests, water areas, arable land, orchards and gardens, vineyards, permanent grassland, built-up areas and other areas. The coefficient was calculated in the period 2002 to 2020. The previous period could not be evaluated as suitable statistical data are not available as certain modifications were made in the monitoring of the area of land use elements in 2000.

The input indicators for the calculation are the area of individual landscape elements (ha) and the coefficients of landscape ecological significance of the elements of the landscape structure Miklós [43]. The most ecological elements of land use are considered to be natural and close to nature elements of land use (forests, water areas, permanent grasslands). On the other hand, built-up and degraded areas are considered to be the least valuable. Areas of land use elements were taken from Statistical Yearbook and the Land Fund of Slovakia [39].

To evaluate the impact of urbanisation on the landscape and its components, we used geographical synthesis and evaluation [45], where landscape ecological issues arising from conflicts of interest between the development of urbanisation and the protection of nature and natural resources were evaluated [46–48]. The syntheses were performed in the GIS environment. The following maps were included in the syntheses and assessment:

- The map of representative geosystems of Slovakia [49], NATURA 2000 map and map of protected areas of Slovakia [50] served as a basis for assessing the impact of urbanisation on ecosystems, including forests and their ecosystem services;
- The map of soil-ecological units of Slovakia, which represents the main soil-climatic units, which are divided in more detail according to the categories of their slope, slope exposure to the cardinal directions, skeletality, soil depth and grain size of the surface

horizon and reflects soil quality based on soil valuation [51]. This map is the basis for assessing the impact of urbanisation on soil resources;

- The map of landscape archetypes [52] and the map of historical structures of agricultural landscape [53]—they were used to assess the impact on the landscape, its type and nature;
- Models of climatic conditions of Slovakia [54–56]. Figure 1 shows the areas with the increased risk of changes in land use in the future in relation to air temperature [54].

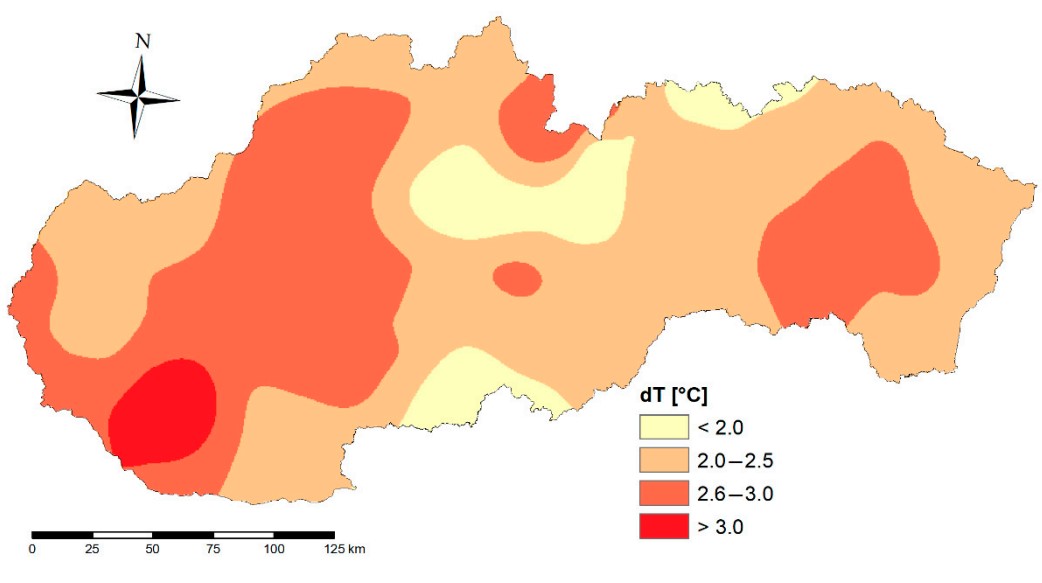

© Slovak Hydrometeorological Institute, 2019

**Figure 1.** Deviations of the average annual air temperature from the normal 1961–1990 in Slovakia in 2018 [54].

To complement the assessment of the impact of urbanisation on the landscape and individual components, an analysis of documents was also performed, especially national and regional conceptual development documents, as well as an analysis of the area in terms of major visual changes in the landscape.

## 3. Results

Slovakia has undergone major transformations changes in its economy over the last 20 years which resulted in increased urbanisation and industrialisation. The market economy conditioned the cancellation of many production facilities, including agricultural cooperatives. New solutions and impulses had to be found for the economic recovery. This issue was solved in the form of the inflow of foreign capital into the territory of Slovakia, which conditioned the construction of new industrial plants and industrial parks which, in terms of urbanisation, have also brought many accompanying changes in land use patterns (e.g., increasing areas with the function of housing, increased interest in the construction of houses in rural settlements located nearby industrial parks). Most of the new industrial buildings were built on the so-called arable land of the agricultural landscape. Old buildings remained abandoned, and the number of brownfields increased (Figure 2). Of the total 74 industrial parks in Slovakia, the most (47) are of the "greenfield type", and 27 are of the "brownfield" type. The total area of all 74 industrial parks is 2858.41 ha. Moreover, 18 industrial parks are registered with an area of less than 5 ha, 28 industrial parks with an area of 5 to 20 ha, 13 industrial parks with an area of 20 to 50 ha and 15 industrial parks with an area of more than 50 ha [43]. According to the structure, most brownfields are in industry (44.4%) and agriculture (20.4%) (Figure 3).

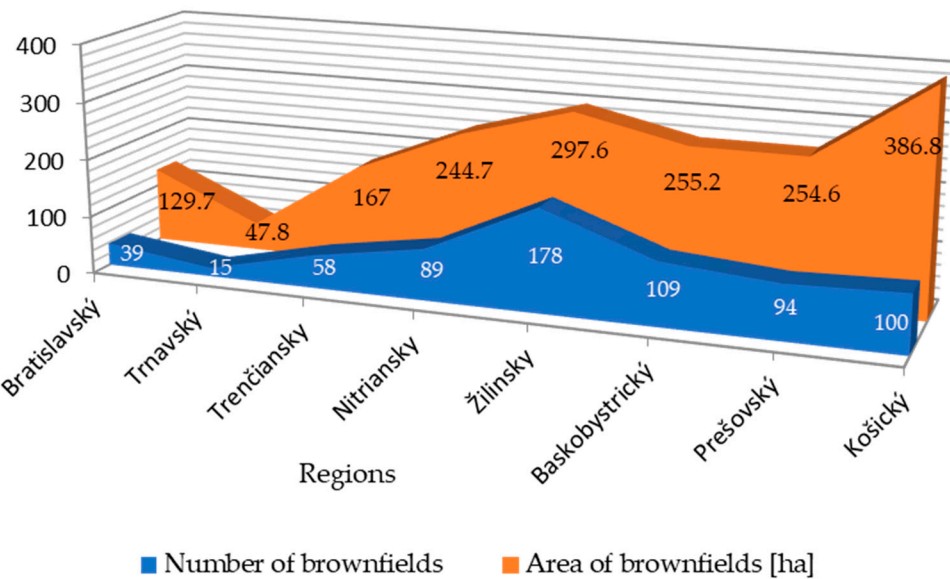

**Figure 2.** Brownfields in regions of Slovakia by 2018 [52,53].

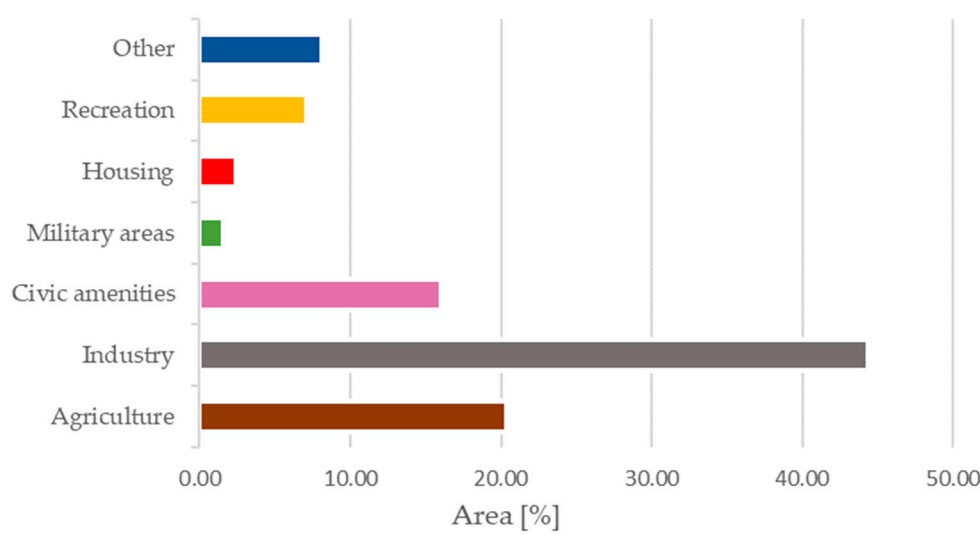

**Figure 3.** Structure of brownfield types in Slovakia by 2009.

The availability of newly created industrial facilities also required the construction of the necessary new technical infrastructure—service facilities, transport corridors, logistics centres which also require certain changes in the landscape structure. Within the settlements, there has been a significant increase in shopping centres. The year 2006 was characterized by the arrival of multinational retail chains. The construction of anthropogenic elements increased the share of built-up areas. From the establishment of the independent Slovak Republic until 2020, the share of built-up areas increased by 110,257 ha, which represents an increase of 85% (Figure 4). The most significant growth of built-up areas was recorded after the division of Czechoslovakia and the establishment of the independent Slovak Republic.

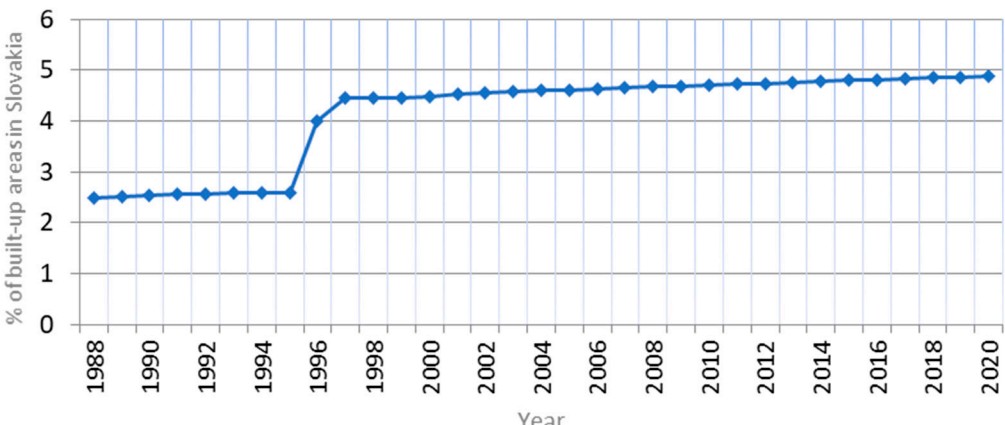

**Figure 4.** Development of built-up areas in Slovakia [23,24].

The construction of new objects related to development was performed only in well-accessible areas, especially in the area of western and central Slovakia where the transport corridors are built. The automotive and electronics industries have become the dominant industries within the industrial sector. Facilities of four car manufacturers—world-famous brands—have been built in Slovakia: Volkswagen in Devínska Nová Ves (west Slovakia), the French PSA near Trnava (west Slovakia), the Korean Kia near Žilina (middle Slovakia) and, since 2016, the Indo-British Jaguar Land Rover near Nitra (west Slovakia). Samsung Electronics in Voderady and Sony located in the Nitra Industrial Park can be considered the flagships of the electronics industry. Regarding to the territory of Slovakia, these companies are disproportionately located in the western part of Slovakia (Figure 5). This is a typical unbalanced distribution of economic interests, which are linked to the requirements of technical infrastructure. Therefore, the best developed areas are in Western and Central Slovakia. Urbanisation in Eastern Slovakia is the least recorded. A degradation of the territory in the form of the abandonment of rural but also urban settlements dominates, the land is uncultivated, brownfields are formed and so on.

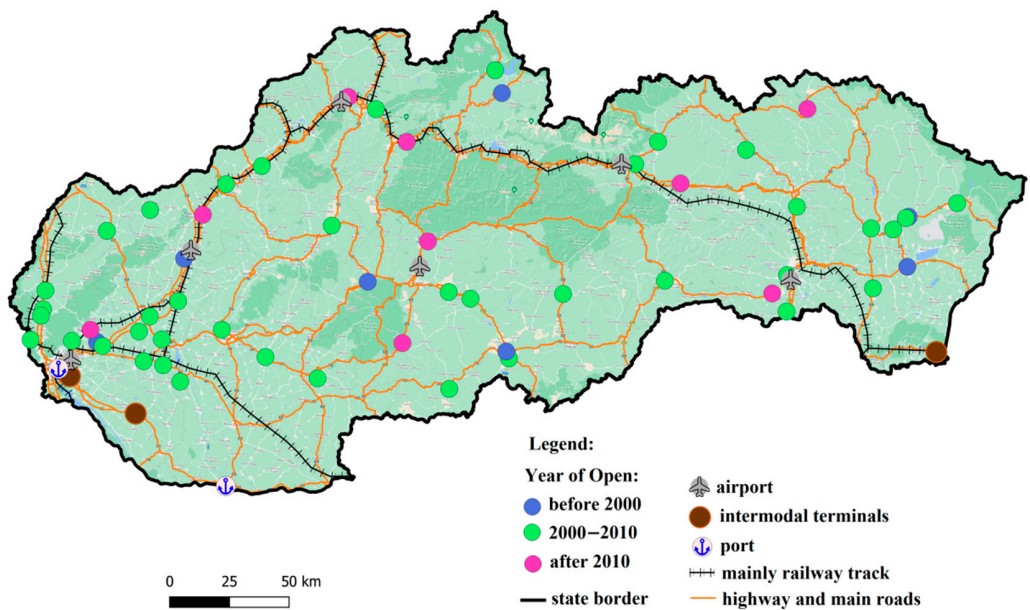

**Figure 5.** Spatial distribution of industrial parks in Slovakia and their classification by the year of open.

The development of urbanisation and industrialisation has caused a significant impact on the natural structure of ecosystems. The negative consequence was not only the area of natural ecosystems but secondarily also the production of pollutants, dust, radiation, etc., which affect the natural development of a lot of habitats. Representative geoecosystems in the areas of lowlands and basins were anthropogenically transformed—especially river floodplains, terraces, proluvial cones, plains, loess boards and hills, polygenic hills or fragmented pediments mostly with original oak or oak-cerium forests. Wetlands, stream-side vegetation, orchards, gardens, vineyards as well as public greenery in settlements are often used for construction [48]. Investor pressures on protected areas are also frequent. For example, the construction of the Samsung industrial unit was performed in collision with the Uľanská mokraď Special Protected Area, and housing construction and the development of new housing estates in the town of Trnava was performed in collision with the Trnavské rybníky protected area. The Šúr National Nature Reserve is very intensively affected by the development of family houses, the motorway and related road connections. Dunajské luhy, the territory of the Natura 2000 network, is directly affected by the motorway bypass of the capital Bratislava. Significant pressures and negative impact on ecosystems due to the development of recreational activities were also recorded in the Tatras Biosphere Reserve and in the Low Tatras National Park (Figure 6). The construction of recreational facilities, especially downhill skiing paths, golf courses and entertainment centres, is not only an intervention in natural ecosystems but also an intervention in the traditional type and nature of the landscape. These new artificial objects represent barriers to territorial systems of ecological stability and cause fragmentation of the landscape (Figure 7). They also contribute to increasing the degree of anthropisation and endangering the spatial ecological stability of the area [43]. As can be seen, the degree of anthropisation increases but the population decreases, so the spatial demands of the population increase (Figure 8).

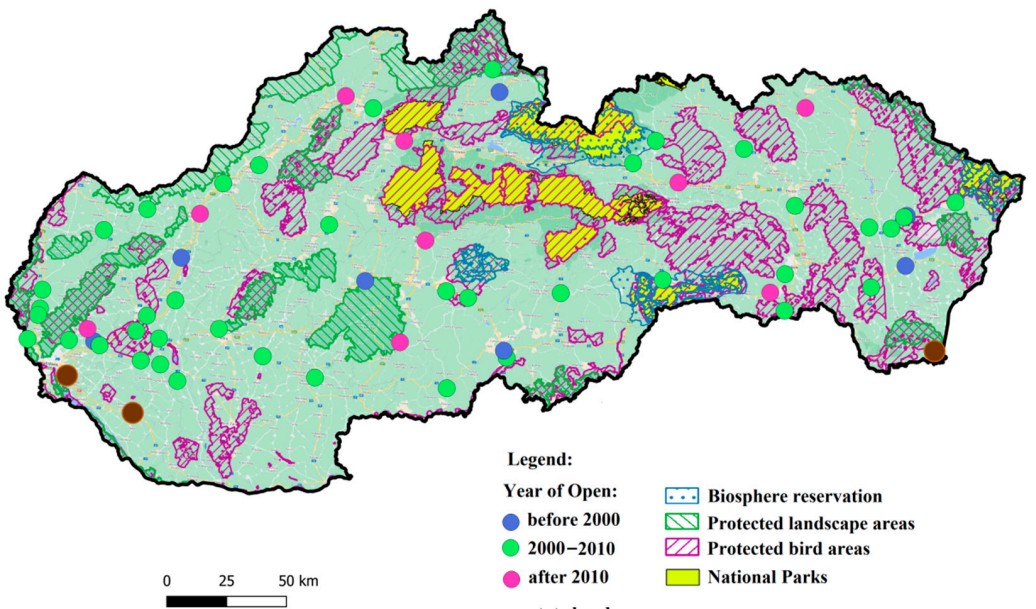

**Figure 6.** Conflict of interests of elements of ecological stability and built-up area, especially industrial parks in Slovakia [37,38].

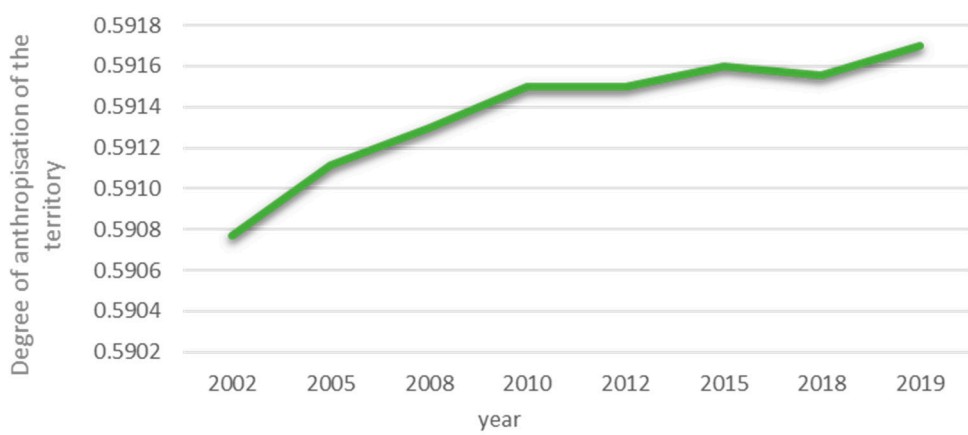

**Figure 7.** Degree of anthropisation of the territory of Slovakia calculated according to Miklós [38] and data from [37,38].

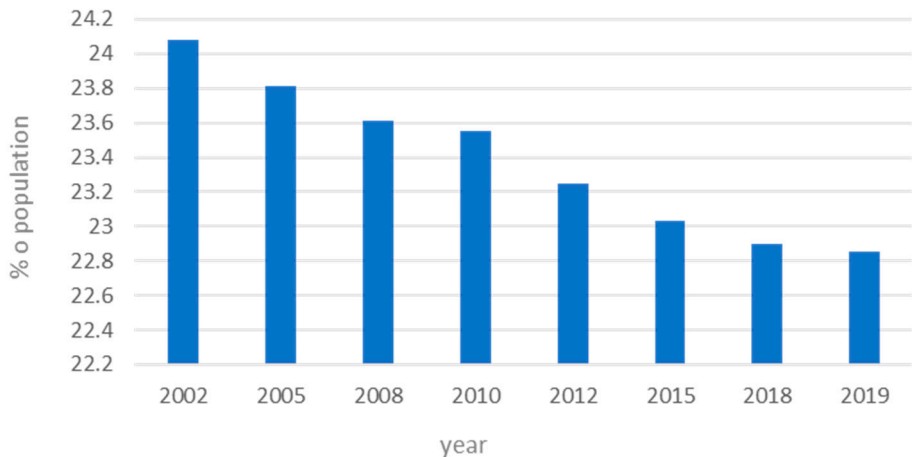

**Figure 8.** Population per built-up area in Slovakia [37,38].

Urbanisation brings a higher proportion of paved areas in settlements that form heat islands. There are only a few settlements in Slovakia that have started to intensively promote and realise the principles of climate change mitigation in their planning processes. Due to the continuing warming of air, based on the outputs of climate models, it is expected that the annual average air temperature in Slovakia should increase by 0.8–0.9 °C in the time horizon up to 2025, by approximately 2.0–2.5 °C by 2050 and by approximately 3.5–4.0 °C in 2100 (Slovak Hydrometeorological Institute). Therefore, it is not desirable to create new heat islands as a result of urbanisation. Significant increases in the daily maximum and minimum air temperatures are expected. On the horizon of 2050, we expect a significant increase in the number of summer and tropical days, but the number of frost days and ice days will decrease. The most important consequence in terms of thermal comfort is the increase in frequency, length and intensity of heat waves, which may occur as early as May and will not be rare until mid-September. A higher number of days with muggy weather is also expected due to the overall increase in the parameters of water content in the atmosphere. A faster onset of warm and dry weather in the spring is expected. In the warm part of the year, the variability of precipitation totals is expected to increase. Apparently, on the one hand, low precipitation (dry) periods are likely to be prolonged and occur more frequently, and on the other hand, short rainy periods abundant in precipitation will occur. On average, precipitation totals will increase slightly over the year. Due to the expected higher air temperature, the evaporation will also increase, which will create conditions for a longer duration of drought, especially in the southern regions of Slovakia. Torrential and intense long-term precipitation is likely to be more frequent and intense

(approximately 7–14% for every 1 °C of warming). Due to the higher temperature and humidity, stronger and more intense storms are expected to occur more frequently. An even bigger problem is and will be the increasing intensity of short-term precipitation, which is reflected in the fact that precipitation is simply more severe. In particular, the intensities of 5 to 180 min of rain increase significantly, which can be explained by the more frequent occurrence of short-term convective precipitation and, conversely, by the rarer occurrence of long-lasting, mostly stratiform precipitation. The larger the proportion of such surface, the largest the share of the precipitation flowing off and the smaller the proportion and quantity of water filtering into the soil. This increases the threat of floods, the pollution of waters and decreases the amount of subsoil water.

In addition to the scope of natural ecosystems, the development of industrialisation and urbanisation also contributes significantly to the threat and degradation of natural resources, especially soil resources. Many industrial as well as residential areas are often built on the highest quality soils in Slovakia. Investments, and therefore land take, are concentrated in regions with the highest quality land. For example, 183 hectares of quality land in Slovakia were seized for the construction of the PSA Peugeot Citroën industrial plant, for the Jaguar Land Rover plant 30 ha, for the Kia plant 166 ha of medium-quality soils and for the Volkswagen plant 200 ha of quality soils. In addition, 130 ha of the highest quality soils were occupied in the Voderady industrial zone. While the construction of industrial buildings and parks was performed mainly on arable land, the construction of residential areas was mainly occupied by orchards, gardens and vineyards, especially in settlements, as it was often the private property of the owners performing the construction. Compared to 1990, the area of vineyards decreased by 65%. The decline in agricultural land by type of land is shown in the following graphs (Figures 9–11). A total of 26,056 ha of agricultural land was occupied for construction in the years 2000 to 2019. Most agricultural land was used for housing construction, construction of recreational facilities and service infrastructure. In the years 2000–2019, 19,493 ha were occupied for these purposes. Significant occupations were also made for the construction of industrial buildings, where, in the years 2000 to 2019, 5511 ha of land were occupied for these purposes, mostly very-high-quality soils—black earth, brown earth, etc. The largest occupations were made in 2006 to 2008 [57]. In terms of protection of quality agricultural land, supranational and national economic interest in the decision-making process was strongly preferred. Large occupations for the construction of residential areas are still realized today. The construction of industrial areas has slowed down after 2010 (Figure 12).

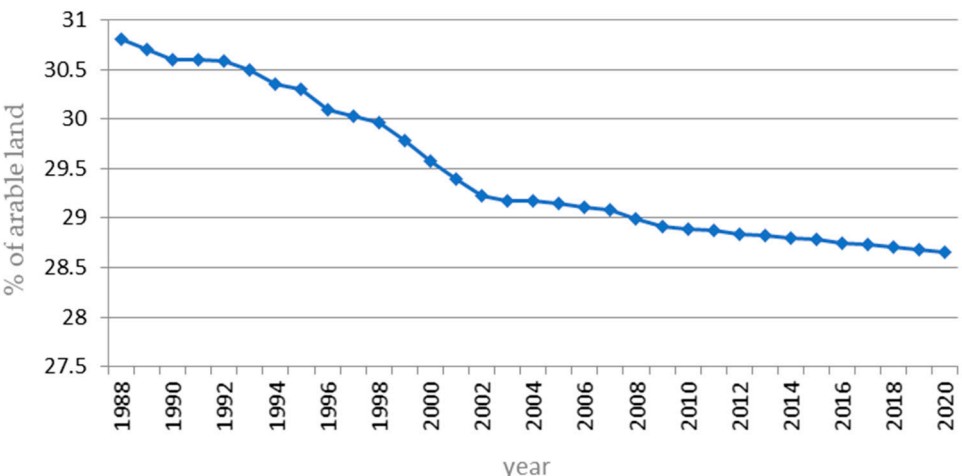

**Figure 9.** Changes in the area of arable land in Slovakia [37,38]. The decline in the area of agricultural land is due a consequence of urbanisation pressure and high level of built-up areas.

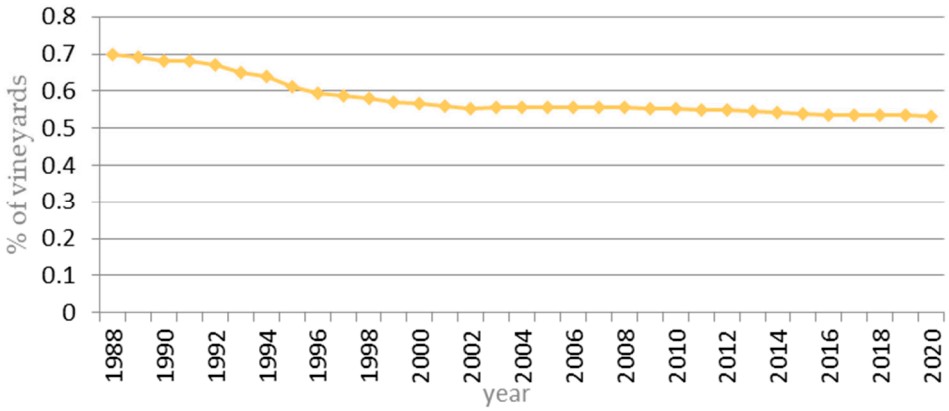

**Figure 10.** Declining trend in the area of vineyards in Slovakia [37,38].

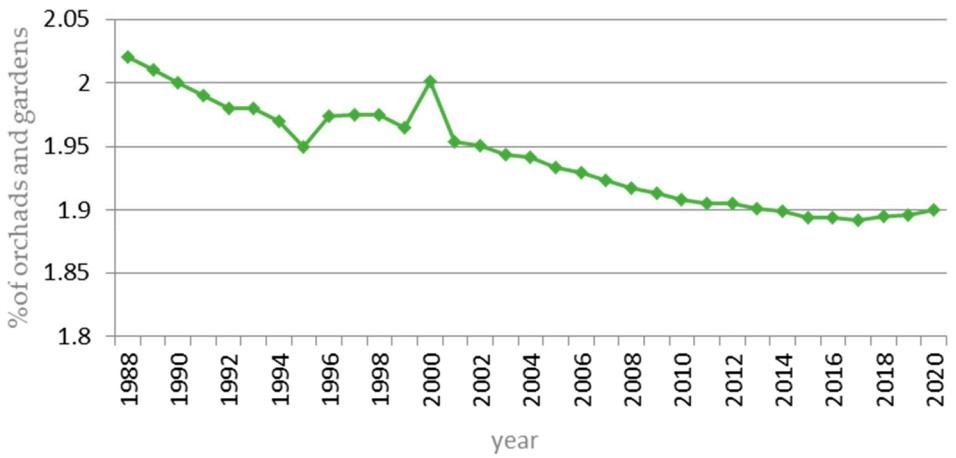

**Figure 11.** Declining trend in the area of orchards and gardens in Slovakia [37,38].

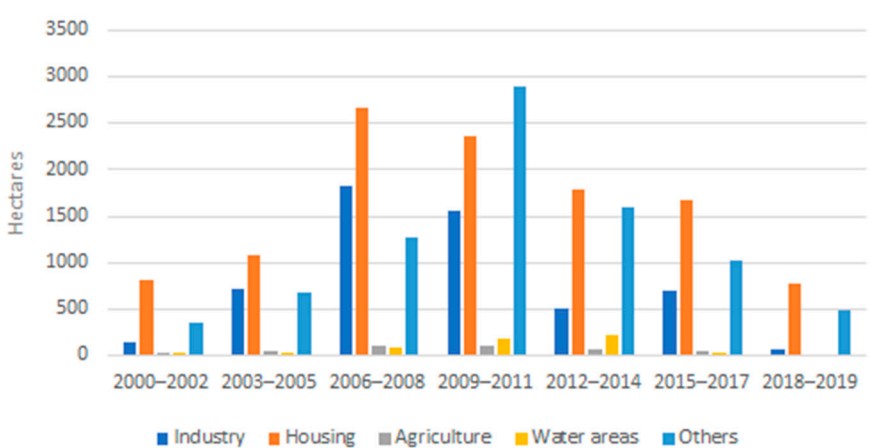

**Figure 12.** Occupancy of arable land in Slovakia for construction [37,38].

The development of urbanisation and suburbanisation has also affected the image and nature of the landscape, especially in the urban functional areas. In the peripheral zones of the cities, industrial zones and zones of shopping centres grew, which gradually began to spread to the surrounding rural area and were a threat to several rare landscape structures. A typical example was the pressure of Bratislava to occupy the picturesque ways of viticultural structures in the Malé Karpaty Mts in Bratislava and surrounding settlements. In the towns themselves, there was a compression of construction, which was

also done at the expense of many "green" areas. Tall buildings atypical of "socialist" towns began to grow in the towns. A lot of Slovak cities are characterized by architectural chaos, which is the result of a mix of socialist and modern styles of construction. *The Post Socialist City* reveals that many cities across Eastern Europe remain dominated by the industrial complexes and panel buildings erected by socialism [25].

The migration of the urban population to rural settlements to more secure environments, more generous living spaces and more individualized housing types was reflected in the expansion of housing construction in rural settlements. In many rural settlements, new residential areas have been built, often not architecturally fitting into the rural nature of the settlement. For the most part, these neighbourhoods have only the nature of "dormitories". Various additional services are also being built in the countryside, such as entertainment centres, golf courses, downhill skiing paths and water parks that do not fit into the image of the country settlements. Gradually, urban elements are being transferred to the rural environment. These processes are subsequently reflected in a change in the demographic structure of the population of rural settlements, a change in lifestyle as well as the creation of a new image of the settlements. In many rural villages, the original architecture, indigenous traditions and indigenous culture began to vanish [58]. The interaction of new settlers and old settlers is problematic in many cases. Immigrants often find it difficult to assimilate into the original settlement community, and so new elements appear in the settlement community—isolation, closure, separation, restriction of communication, egoism, preference for their own local interests, loss of interest in public affairs, etc., which is reflected in changing their lifestyle and consequently on the quality of life of the settlement community. As in other countries, people with a high social status move to rural settlements much more [59].

On the other hand, rural settlements in marginal areas without economic impulses are declining more and more. Due to the lack of job opportunities, the population of younger age groups is leaving for work, and the older population does not have sufficient financial or technical resources for management, investment in business activities or investment in housing development. The displacement of settlements causes significant ageing and their abandonment. Many marginal settlements often turn into cottage settlements. This unfavourable ratio of the ageing index (higher than 100) occurs in almost two-thirds of rural regions, i.e., in 45 districts. These are mainly districts of western Slovakia with scattered settlements (Myjava, Nové Mesto nad Váhom, Piešťany) and the districts of Medzilaborce, Komárno, Nové Zámky and Levice (in the range of 125–149). In the observed period, the representation of the pre-productive population in these rural regions decreased significantly, which is also related to their significant reduction in natural increase [60].

## 4. Discussion

The process of transformation in Slovakia, as well as in other countries of Eastern Europe, significantly influenced the development of urbanisation [61]. The transformation in this region is similar to the urbanisation processes of Western Europe but is significantly delayed compared to Western European countries [62]. The delay was caused by the centralisation and regulation of the development of urbanisation in the pre-transformation period. The construction of standardized uniform prefabricated housing units was preferred. Serial settlements were massively built in towns, which gave them the typical image of a socialist town. In rural municipalities, a policy of the centralisation of settlements was applied where the preference for the development of rural settlements was determined. A part of the settlements was developed dynamically and part was destined for extinction. The turnaround took place in 1989 after the onset of the market economy, when business development took place, the market was opened to foreign capital, ownership relations changed and restitution and privatisation took place there, too.

The decentralisation of power has affected residential development both positively and negatively. The gap between the individual municipalities and the regions has widened significantly, and the marginalisation of certain spatial units has become more pronounced.

Location has become a basic development factor. Rural settlements close to large industrial and urban centres, as well as settlements with a favourable transport location, began to develop rapidly. New residential areas are being built there; entertainment centres, golf courses, downhill skiing paths, water parks and similar sports facilities are being built that are not typical of the socialist era; diverse production facilities are located there, which, in turn, is reflected in the demographic and economic development; settlements have better service networks; and we can see the overall development of municipalities. Many rural settlements act as residential satellites for the surrounding towns.

As a result of this situation, rural settlements are gradually losing their primary agricultural function. The development of the agricultural land market has meant that the highest quality land is often sold for the construction of family houses, logistics centres, recreational and various other technical facilities. In this way, new, often disturbing, elements enter the rural environment. On the so-called green meadows, new residential areas, luxury recreational complexes and production facilities are being built, often not respecting the environmental and aesthetic requirements of the rural environment. Similarly, location has played an important role in the development of industrialisation. From Figure 5, the connection between the location of industrial parks and transport corridors can be seen. Most industrial parks were built in connection with the highway D1, in the part of Bratislava–Žilina and on the expressway Bratislava–Nitra. Many sections of these roads have been built on arable land, and many habitats have disappeared. On the other hand, as a contrast, we find the remains of old abandoned buildings of former cooperatives, state property, industrial enterprises, but also abandoned residential and farm buildings. There were several landscape ecological issues associated with the development of industrialisation and urbanisation, which were also identified in other countries and were described by several authors [17,18,20,22,24,59,63–68]. The most important environmental problems associated with the transformation process in Slovakia are the following:

- Irreversible land use change—urbanisation changes the natural landscape into semi-natural and artificial, increases the degree of anthropisation, reduces the spatial ecological stability of the area, causes landscape fragmentation, threatens the migratory movement of biota and threatens biodiversity [69,70];
- Persistent pressures of investors on the occupation of natural resources—construction is mostly performed on agricultural and forest land, which is associated with the scope of natural resources, natural ecosystems, rare historical landscape structures—vineyards (also historical vineyards with the highest landscape value), orchards and vegetation areas. For Slovakia, the area of the highest quality soils for development is also typical, especially for the construction of industrial parks and buildings; occupancy, an increase in built-up areas and impermeable surfaces worsen the adaptation landscape balance;
- Change in the use of ecosystem services—the process of urbanisation negatively affects the ecosystem services [12,71–73], including food production, gene pool protection, regulation natural processes, endangering natural aesthetic values; these do not create the right conditions for the landscape to adapt to the effects of a changing climate;
- Changes in the image of the villages and rural country—the growth of new residential areas inappropriate for rural areas (unsuitable architectonic design of constructions), loss of the nature of rural gardens near houses in rural settlements, inappropriate landscaping around the rural houses (paved areas, inappropriate plant species). These processes are also reflected in the reduction of the aesthetic value of the landscape;
- Lifestyle change in rural settlements—population growth in peri-urban areas, transfer of the elements of an urban lifestyle into the countryside, decline in community life, increasing manifestations of separation, egoism, etc. [66];
- Negative impact on climate change—increasing share of built-up areas reduces seepage, accelerates water run-off, increasing risk of floods as well as overheating of surfaces and the creation of heat islands [13];

- Increase in traffic intensity and associated negative effects—the relocation of the population to the rural environment caused an increase in transport, especially daily attendance for work, which subsequently caused increased noise, dust but especially increased production of traffic fumes.

Although many environmental impacts are directly observable, they are much more difficult to measure and evaluate their effects [74]. The development of urbanisation must be assessed comprehensively, taking into account all the negative effects. Not a single negative impact can be neglected. The basic tool for the elimination and especially prevention of environmental problems related to the development of urbanisation is the implementation of landscape ecological regulations into spatial planning processes. Although in Slovakia, according to the valid Building Act, the landscape ecological plan is a mandatory part of the zoning plan, its position is problematic and insufficient. The requirements of a landscape's adaptation to climate change are not yet mandatory in spatial planning. There are no methodologies for processing modern land use plans, in which relevant spatial data are projected. The processing of planning process documentation as well as their strategic assessment is often formal, focusing only on mandatory compliance with legal standards and directives. It is not precisely defined how the regulations should be reflected in spatial planning. The act defines that the spatial plan is a mandatory part of surveys and analyses, and so it often happens that the land plan is processed but no longer enters into binding regulations for spatial development. Slovakia has very well-developed planning methodologies, but the processing of these planning documents is often very formal, focused only on mandatory compliance with conventions and legal standards. According to mayors, the landscape-ecological documentation is very often considered to be unnecessary and insignificant, and it is a quite demanding and complicated processing of planning documents. Environmental legislation is sometimes considered an obstacle to development. On the other hand, some legislations are real obstacles in landscape development, e.g., very rigorous and lengthy environmental impact assessment processes with requirements for the research and monitoring of landscape components. In most municipalities, the need to build investment plans is considered a priority. In the real practice of implementing the planning documents, owners and 'investors' pressures often cause changes to the proposed and planned activities in urban plans. The problem is also the complexity of the methodological procedure for processing documentation—the recommended methodology of the landscape ecological plan by the Ministry of the Environment of the Slovak Republic [75] is built on an interdisciplinary basis. It is based on the concept of "landscape as a geosystem". Interdisciplinary processes emphasize the integration of knowledge about the country [76]. The interdisciplinary approach is generally well declared, but in real life, it is applied less frequently, as it is a very complex process. The success of this process depends on the ability to integrate all types of knowledge available for a specific setting. It is believed that this co-production of knowledge depends strongly on a shared assumption amongst all participants that all knowledge is equally relevant to the process [77,78]. Transdisciplinarity excels in breaking out of the academic context and finding ways to use other knowledge bases, such as those provided by non-academics, stakeholders and practitioners [79].

## 5. Conclusions

The development of urbanisation in Slovakia after the transformation took place without effective control and planning, which caused the occurrence of several landscape, ecological and environmental issues. Many natural ecosystems have been occupied and degraded, the shares of elements of green infrastructure have been reduced, the degree of anthropisation of the area has increased and a considerable amount of the best soils have been taken up. The use of natural resources as well as the growth of anthropisation is growing despite the fact that the population is not increasing. It is assumed that these trends will continue. The reason is probably that lifestyle and housing demands are constantly increasing. The demand for new areas for the construction of new housing units is constantly increasing, the share of natural and semi-natural ecosystems is gradually

decreasing and the share of areas with an impermeable surface is increasing. The reasons that cause the growth of built-up areas have not yet been sufficiently investigated, and they need to be given increased attention. If we want to eliminate these negative effects in the future, it is necessary to ensure the regulation of urban development in accordance with the landscape and ecological conditions [80]—to control the area of the best soils, to control the occupation of natural ecosystems as well as the limits for construction density, etc. In the future, the environmental effects of growth would need to be monetized. To this end, it is necessary to develop methods for the economic assessment of these negative impacts and to introduce construction fees that also include environmental costs. This situation calls for a focused and multi-dimensional planning approach to peri-urban areas that includes ecological value, socio-economic issues, perceived quality of life and the nature of the planning system [81–83].

Evaluating the impact of urbanisation on the landscape and the environment requires new methods and new approaches that consider not only the complexity of urban dynamics but also the impacts and responses to these dynamics [84]. Future research should also focus on assessing the socio-economic factors and the impact of lifestyles on urban development.

If we want to secure natural resources for future generations, it is appropriate to set spatial limits for development and also to set limits for the use of natural resources on the basis of their qualitative and quantitative properties. This needs to be applied in strategic development documents, municipalities and regions, economic and social development plans and zoning plans. However, their strict adherence is a necessary condition, which is not always consistently implemented at present.

**Author Contributions:** Resources, F.P.; Writing–original draft, Z.I.; Writing–review & editing, E.P. All authors have read and agreed to the published version of the manuscript.

**Funding:** This research was funded by Scientific Grant Agency of the Ministry of Education, Science, Research and Sports of the Slovak Republic and the Slovak Academy of Sciences, grant number VEGA 1/0658/19 and Slovak Research and Development Agency grant, number APVV-17-0377.

**Acknowledgments:** The work was supported by the Slovak National Grant Agency, by projects: Ecosystem approaches to assessing of anthropogenic changed territories according to selected indicating groups of species and Evaluation of modern changes and development trends of the agricultural landscape of Slovakia.

**Conflicts of Interest:** The authors declare no conflict of interest.

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
