# Peer review of "The Impacts of Urbanisation on Landscape and Environment: The Case of Slovakia"

_sustainability, doi:10.3390/su14010060_

Round 1

Reviewer 1 Report

The subject of the article seems interesting. However, after reading the manuscript, the following points were raised. 

- there is no literature review on the problem in question and other possible methods than the one proposed by the authors; 

- line 171: "... n is the number of elements in the area." - there is no variable "n" in the formula in line 166; 

- formula in line 166: the values for the variables used in the analysis are missing and the calculation results are missing; 

- line 192 and Figure 1: "Models of development of climatic and bio-climatic conditions of Slovakia [39-41] (Fig. 1)." - there is no explanation as to what is shown in the map in Fig. 1. The descriptions in the legend on the map are too general; 

- Fig.2: missing data for "Bratislavsky" - is it supposed to be like that, is it a mistake? 

- Fig.3: Caption "...(area in %)" - there is no value in the chart; 

- Fig.4: no description of the OY axis; 

- Fig.9: no description of the OY axis; 

- Fig.10: no description of the OY axis; 

- Fig.11: Brak opisu osi OY;

- Fig.12: Brak opisu osi OY;

- section "4. Discussion": too general considerations interspersed with elements of a literature review; a literature review should be placed at the beginning of the article to justify the problem selection and present state-of-art; 

- section "5. Conclusions": the conclusions are very general and loosely related to the results of the analysis. 

GENERAL:
- The content of the manuscript is scientifically very poor;
- It is recommended to increase the scientific aspect of the issue under discussion;
- it is recommended to specify the purpose of the research more precisely and indicate what is new about the presented results. 

Author Response

The authors' notes for the reviewer are summarized in the attached file.

Reviewer 2 Report

The paper focuses on the evaluation of the environmental impacts of urbanization in Slovakia. This is a solid paper devoted to the pertinent issue of assessment of changing landscapes and nexus with urbanization, planning and development. The layout of the paper and the research design is correct. Some small changes can improve the paper for international readers:

1. the title: I propose to change it to better enhance the content: The impacts of urbanisation on landscape and environment: the case of Slovakia.

2. In the conclusions many bibliographical citations of other scholars are included. In the conclusions it is preferable to focus on the main results of the paper and to bring into the discussions the evidence from other studies (522-523, 530, 538, 546-547). Therefore, the policy and planning implications evidenced by the authors as well as the role of the decision making process (rural development programs, and agricultural policies) could be better highlighted.

Author Response

(The authors gave the same response as above.)

Reviewer 3 Report

Review of  “ The Environmental Impacts of Urbanisation – Slovakia as an Example”

Manuscript ID: sustainability-1425335

The paper concerns the issue of environmental changes associated with the development of urbanization in Slovakia. The paper is of a general nature, mainly due to very (too?) broadminded approach to the subject.  It is rather general overview of various effects of the political and economic transformations that occurred in the years 1998-2020 than good evaluation of ecological problems established in a scientific manner.  

General comments

Comment 1: Abstract

The abstract mainly contains historical background of urbanization process in Slovakia. There is no description of methods that were applied in the study. Furthermore, it is not stated what are the main results of the study.

Comment 2: Introduction - 120-137

In the introduction the Authors formulated four hypotheses. However, they refer to commonly known facts and therefore do not have to be proven. Since the study concerns urbanization in Slovakia, I suggest to formulate research hypotheses that refer to this specific spatial context.

Comment 3s:  Materials and Methods -  Lines 161-164

One of the Authors’ contributions to the study is an assessment of ecological quality of the area.  Several coefficients describing the degree of anthropization are mentioned. All of them have been developed by Slovak authors. The literature review should also include some foreign publications. Short information regarding main differences between the described methods should be presented. Especially that  it is not clear why the coefficient of Miklos was chosen to be used in the study. Furthermore, I could not find the results of this assessment.

Comment 4:  Materials and Methods - Lines172-174

“The input indicators for the calculation are the area of individual landscape elements  (ha) and the coefficients of landscape ecological significance of the elements of the landscape structure” – this is unclear. What do you mean by elements of landscape structure? What is the source of data? The method to calculate the coefficient of landscape ecological significance should also be properly described.

Comment 5:  Materials and Methods -  Lines 175-179

This part of Material and Methods requires additional explanation. It is clear that in order to evaluate various ecological conflicts the results of other evaluations of natural resources in Slovakia were adopted in the study. However, we do not know how the assessment of impacts of urbanization on soils, landscape, ecosystems, bio-climate conditions  etc.  was conducted in the presented study. Moreover, this kind of input data (i.e. maps) allows for analysis of environmental impacts in quantitative and qualitative  manner  and present their spatial extent. Unfortunately, the presented results of the analysis are rather mostly descriptive and rather general.

Comment 6:  Materials and Methods - Lines 197-201

The part concerning additional sociological survey is too general and requires additional and more detailed explanation. Otherwise, the methods in their current state are not viable for use in other research and the obtained result are difficult to justify in terms of  scientific soundness.

Comment 7:  Results Lines - 297-298

How was the degree of anthropization calculated?

Comment 8:  Results - Lines 360-362

This part concerns designation of agricultural land for construction purposes.  According to the Authors, large area of good quality agricultural land was designated for residential areas and industrial plants. Does it mean that there was no legislation protecting good quality soils against conversion into building sites? I suggest adding some additional comments in this field.  

Comment 9:  Results - Lines 384-395

I recommend to exclude sociological analysis from this chapter and focus on the issues that are strictly associated with the title and objectives of the study.

Comment 10:  Discussion

The discussion section should be focused on your own findings, especialy those worth discussing in relation to other studies. Please do not present new facts or problems that were not presented in the results or cannot be deduced from your results. (e.g. negative effects of increase in traffic intensity, ecosystem services).

Comment 11. Discussion and Summery

It is common practice to refer to hypotheses formulated in the Introduction. Please consider rewriting the hypotheses and then try to refer to them according to the findings of the study.

Comment 12 : Language

A careful proofreading is needed, as there are many unclear sentences and grammatical errors.

Specific comments

Comment 1: Line 206

The chapter starts with “In addition to socio-economic change..”   This appears to be a continuation of some previous analysis rather than a beginning of a new chapter..

Comment 2: Lines 223-224 Fig. 2

I do not agree that brownfield can be divided into: greenfield brownfield and greenfield/brownfield. What is the source of this kind of division? Perhaps you should define this term.  What is the source of data and to which year do they relate to? A map with the location of the regions listed in Fig. 2 is missing.

Comment 3: Line 226 Fig. 3

The figure presents the brownfields types in Slovakia in the year 2000 rather than “ by 2020”.   Source of data is missing.

Comment 4: Line 329 Fig. 6

Title of the Fig. 6  is not relevant to the contents of the map. Instead of degree of enthronization, the figure presents distribution of industrial parks and various types of nature conservation areas.

Date 25th. of October2021

Author Response

(The authors gave the same response as above.)

Reviewer 4 Report

An interesting subject, applause for the Authors for taking up an important and very topical research topic. However, I have a few remarks regarding the structure, arguments and conclusions resulting from the conducted research, which are discussed below.

Abstract - no clearly defined goal of the work. No information about test methods. Very general description indicating the known fact that "The development of urbanization is linked to qualitative and quantitative changes of the landscape and its components". Some well-known information such as “Independent Slovakia belongs to young European states. The communist period lasted from 1948 to 1989 in the different political-administrative forms. Character of landscape was dependent on centrally managed economy. Slovakia joined the European Union in 2004 and this milestone had also enormous effects for landscape changes. The transformation of central planning into a market economy was the basis of these changes, which conditioned following strong pressure of investors on the landscape, construction of technological parks, shopping and logistics centers, transport infrastructure, but also construction of residential complexes, etc. " The abstract also lacks information on the results of the research undertaken.

Rethink key words. You practically not used indicated key words (in the form as it is written in the key words) in the text. “Landscape changes” only 2 times inside the text - in the abstract and key words; "Landscape character" only 1 time in the text - in the key words. The same "anthropization of landscape"; "Degradation of natural resources" 2 times; “Environmental problems” - 4 times (1 in the key words, 2 times in the abstract, only 1 time inside the text).

Introduction

Part “Independent Slovakia belongs to young European states. The communist period lasted from 1948 to 1989 in the different political-administrative forms. The nature of the landscape was dependent on centrally managed economy" (line 38-40) practically copy from abstract (or abstract copy of introduction) (see abstract, line 19-21).

All four given hypotheses (line 121-133) have already been posed in many works, verified not only in Central European countries but also in many other rules. With all due respect to the work of the Authors, it is difficult to evaluate them as an original contribution to science. It is possible that I am wrong, but the research and scientific verification of H (1) (line 122-126) seems unnecessary to me, for H (2) line (127-128) “The development of urbanization and suburbanization visually changes the image and nature of the traditional landscape” is obvious. The changes in the second half of the sentence are a logical consequence of the development and urbanization in the first part of this statement. No need to prove it, in my opinion. If we have "development of smt." Then the  consequence is "changes".

Material, methods

There is no explanation why 2000 and 2018 were selected for comparison? Why not 2004 (Slovakia in UE), or why not 1990? The works cited in the text [20; 21] refer rather to the data base system, and do not justify the choice of the year. Unless the justification is that we have data from that period. Please explain.

Result & discussion

The Result part lacks a clear division of industrialization and urbanization processes. The fact is that they are related to each other, but the title of the work suggests a greater emphasis on urbanization processes than on industrial ones (line 210-277). Clearly visible no distinction in the sentence (line 277-278) "The development of urbanization and industrialization has caused a significant impact on the natural structure of ecosystems." It seems that the authors investigate the influence of both processes, and yet in the title, goal and research hypotheses, they indicated only urbanization.

The sentence "The construction of anthropogenic objects related to development was performed mainly in locally suitable areas, such as lowlands, basins, foothills and in well-accessible areas" (line 240-241) raises the question of structures that are not anthropogenic. So is their location information. And why would they be located in hard-to-reach terrain?

Figure 6 shows landscape parks, protected areas, and industrial parks - in my opinion not Figure 6. Degree of anthropization of the territory of Slovakia [23-24].

Fig. 7 - the same title like fig. 6.

Fig. 8. Show, that there is less% of population on the build-up area?

It is not clear for me to move in the same paragraph from "anthropization ..." (line 297-298) to "Due to the continuing warming of air ..." (line 299 and next).

Fig. 9 - how do you understand the data in % on the vertical axis? So a decrease of about 6.5%?

Line 359: "black earth, black earth"? Maybe more clearly. Please explain or change.

I do not understand why there is no presentation of the survey results or document analysis as a result, compare the fragment of the methods: "In order to complete the assessment of the impact of urbanization on the landscape and individual components, the analysis of documents was also made, in particular, national and regional concept documents for land development and land analysis for major visual changes to the landscape. In order to assess the issues and possibilities of regulating the development of urbanization, we conducted a sociological survey - a survey which was conducted together with representatives of municipalities in the functional urban area of ​​Trnava. 25 mayors from this functional urban area participated in the survey. " (line 194-201). There are no references to these studies in the text or they are too weakly emphasized.

The given conclusions are obvious, known and are in fact a repetition. It seems to me that some of the conclusions do not result from the given result, but are general, as the authors themselves write (line 444-478).

Literature

Please check the literature, e.g. in [47] (line 642-643) there is no DOI number (should be https://doi.org/10.5937/gp24-25543)

Perhaps the following works or same parts of them would be helpful in the analysis:

Pazur, R., Bollinger J., Enhanced land use datasets and future scenarios of land change for Slovakia, Data in Brief 14, (2017), 483–488484

Robert Pazúr, Janine Bolliger, Land changes in Slovakia: Past processes and future directions, Applied Geography, 85, (2017), 163-175, https://doi.org/10.1016/j.apgeog.2017.05.009

Author Response

(The authors gave the same response as above.)

Round 2

Reviewer 1 Report

The explanations, changes and extensions introduced by the Authors to the manuscript are satisfactory. 

Author Response

The explanations, changes and extensions introduced by the Authors to the manuscript are satisfactory.

Thank you for your acceptance of our corrections.

Reviewer 3 Report

Second review of  “The impacts of urbanisation on landscape and environment: the case of Slovakia”

Manuscript ID: sustainability-1425335

I appreciate that some of my comments were approved by the Authors. The manuscript has been redrafted and improved. However, I would like to provide some additional suggestions .

Keywords:  I suggest to erase “change” – to general,

Lines  81-83

At present in Europe, (…) the literature indicates (…) “ – you cannot describe and analyse the present state by referring to the articles that were published  in 2002, 2003, 2005 and 2011.

 Lines 117-118

I am not convinced that “ The issue of assessing the environmental impacts of urban development in European towns is less elaborated” and “  Much more American authors evaluate the environmental problems associated with the development of urbanization” Especially that in case of “American” studies the authors refer to the articles  from 2001.

Lines 144 -146

“ creation of brownfields of new anthropogenic objects on “green area” – I do not understand this part  - perhaps it is a matter of English proofreading

Lines 151-153 – Hypothesis

I  still thing that this is not a hypothesis that needs to be proven.

Lines  281-258; Figures  2 and 3

Statistical data concerning brownfields are from 2009 and cannot be used to describe existing structure of brownfields.

2021.11.26

Author Response

Keywords:  I suggest to erase “change” – to general, - accepted, the word “change” has been deleted

Lines  81-83

At present in Europe, (…) the literature indicates (…) “ – you cannot describe and analyse the present state by referring to the articles that were published  in 2002, 2003, 2005 and 2011. – accepted; the authors wanted to show that this is a topic that has been receiving attention for a long time by other authors; the sources were added

 Lines 117-118

I am not convinced that “ The issue of assessing the environmental impacts of urban development in European towns is less elaborated” and “  Much more American authors evaluate the environmental problems associated with the development of urbanization” Especially that in case of “American” studies the authors refer to the articles  from 2001. - the sources were added

Lines 144 -146

“ creation of brownfields of new anthropogenic objects on “green area” – I do not understand this part  - perhaps it is a matter of English proofreading – the text were corrected „...creation of brownfields of new anthropogenic objects built on areas which were originally arable land or vegetation areas.

Lines 151-153 – Hypothesis

I  still thing that this is not a hypothesis that needs to be proven. – the degree of anthropization is a variable indicator, so it is important to verify it over time and examine the direction in which land changes are evolving (e.g. in terms of environmental and ecological quality of the environment, etc.); the results of a landscape changes can take a variety of forms, and the results of research can be surprising; we found the similar hypotheses in some published scientific articles

Lines  281-258; Figures  2 and 3

Statistical data concerning brownfields are from 2009 and cannot be used to describe existing structure of brownfields. – an error occurred, we were originally also based on the work from 2018, we did not list it in the text; the error is corrected in the manuscript and the literature is supplemented

Thank you very much for a detailed reading of our manuscript, for your time and all your inspiring comments and remarks. We appreciate it very much.

Reviewer 4 Report

Dear Authors, I am glad that I was able to help improve a large part of the text. In my opinion, the version presented after the amendments is more consistent and transparent. One small detail, please re-check the number of phrases used in the text "anthropization of landscape", which you give in key words.
I am sending greeting and I wish you good luck.

Author Response

Dear Authors,

I am glad that I was able to help improve a large part of the text. In my opinion, the version presented after the amendments is more consistent and transparent. One small detail, please re-check the number of phrases used in the text "anthropization of landscape", which you give in key words.  – accepted, the word “landscape” has been deleted.

Thank you very much for all the comments and remarks on the manuscript. We appreciate you taking the time to write review.